# DSPart: A Large-scale Diffusion-generated Synthetic Dataset with Annotations from 3D Parts

## Abstract

Object parts provide representations that enable a detailed and interpretable understanding of object structures, making part recognition crucial for various real-world applications. However, acquiring pixel-level part annotations is both expensive and time-consuming. Rendering 3D object models with their 3D part annotations is a promising solution since it allows the generation of unlimited synthetic data samples with precise 3D control and accurate part segmentation masks. Nevertheless, these synthetic datasets suffer from a lack of realism, resulting in large domain gaps. In this paper, we present a large-scale realistic synthetic dataset with part annotations, namely Diffusion-generated Synthetic Parts (DSPart), for both rigid objects and animals. For images in DSPart, we obtain 2D part masks from 3D part annotations by leveraging recent advances in diffusion models with 3D control. In addition to offering more diverse and realistic textures, prior knowledge of diffusion models enables the object to exhibit more physically realistic interactions with the ground plane and other spatial contexts. We annotate $475$ representative shape instances from $50$ object categories for DSPart-Rigid and use $3,065$ high-quality SMAL models fitted poses from $40$ animal categories for DSPart-Animal. Experimental results demonstrate the potential of our dataset in training robust part segmentation models, effectively bridging the gap between synthetic and real-world data.

## 1 Introduction

Important contributions in the field of cognitive psychology have evidenced that human perception of objects is strongly based on part decomposition (Hoffman & Richards, 1984; Biederman, 1987; Ross & Zemel, 2006) and have inspired the usage of part-based methods for computer vision. Focusing on parts or their representations, followed by validation of their geometric configurations, offers many advantages. For example, based on parts, a large number of geometric variations of highly articulated objects (*e.g.*, animals) can be represented and learned holistically (Ross & Zemel, 2006). Part-based models have also been shown to be relatively robust to partial occlusion (Kaushik et al., 2024b; Kortylewski et al., 2020) and allow few/zero-shot knowledge transfer to novel objects (Gölcü & Gilbert, 2009; He et al., 2023b). The visual characteristics of specific object parts exhibit less variability under changes in pose (and other nuisance factors) compared to the overall appearance of the entire object (Ross & Zemel, 2006; Kaushik et al., 2024b;a) which enables better model robustness (Sitawarin et al., 2022; Li et al., 2023; He et al., 2023a; Xie et al., 2024; Zhang et al., 2024). These advantages make part recognition important in real-world applications, such as robotics (Aleotti & Caselli, 2012; Nadeau et al., 2023) and action recognition (Zhao et al., 2017; Wang et al., 2012).

However, the predominant focus within the community in the big data era has been on addressing tasks at the object level, with minimal attention given to intermediate part representations. This shortage is mainly due to the lack of datasets with pixel-level part annotations across general categories. Most existing part datasets focus on a small number of object categories, such as humans (Li et al., 2020; Gong et al., 2017; Li et al., 2017; Yang et al., 2019) and cars (Dinesh Reddy et al., 2018; Song et al., 2019). Although some recent works have presented relatively large-scale part datasets (Chen et al., 2014; He et al., 2022), the absolute number of images annotated in these datasets still falls far behind those with pixel-level annotations on general object-level categories. For example, both

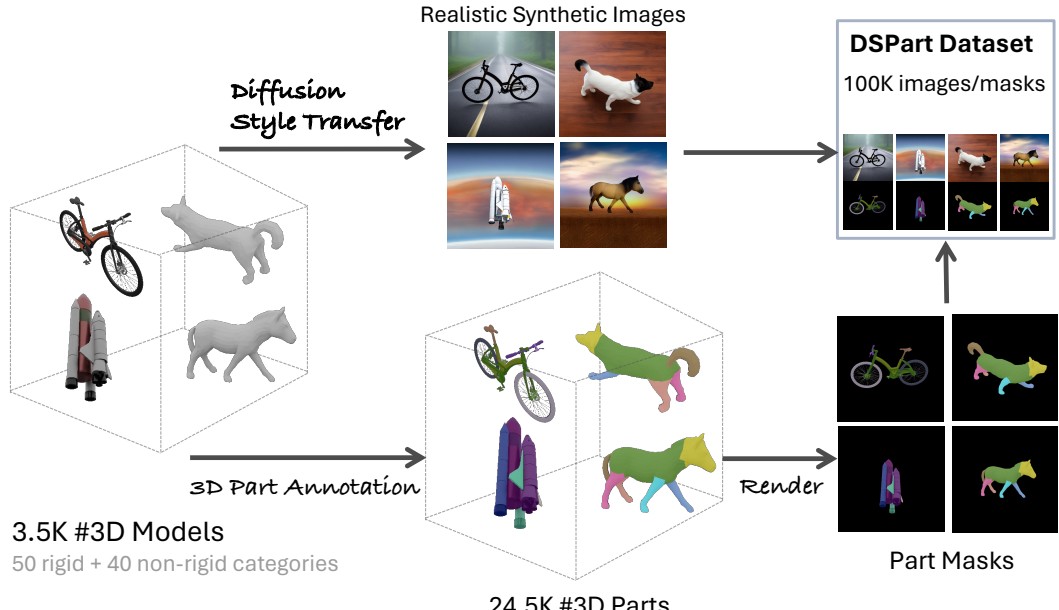

**Figure 1: Overview of the DSPart framework**. Firstly, we define 3.5K 3D CAD models with 50 rigid and 40 non-rigid categories. Secondly, following the 3D-Diffusion Style Transfer(3D-DST) (Ma et al., 2024) synthesis pipeline, we generate 100K realistic synthetic images with 3D pose ground truth (*i.e.* rotation matrix) from our 3D models. Thirdly, we annotate 24.5K 3D parts on our models, rendering 100K accurate part masks aligned with 3D pose ground truth. The 100K rendered part masks are associated with the 100K synthetic images, provided by the same associated 3D pose ground truth. The DSPart framework signifies a paradigm shift in the curation of part datasets by integrating scalable 3D rendering with diffusion-based realistic image synthesis.

Pascal-Part (Chen et al., 2014) and PartImageNet (He et al., 2022) include only around 20K images with part annotations, whereas COCO (Lin et al., 2014) includes 205K annotated images at the object level. This disparity highlights the need for more extensive and diverse part-annotated datasets.

The aforementioned disparity exists due to the complicated and expensive nature of creating accurate large-scale pixel-level part annotations. Using foundation models such as SAM (Kirillov et al., 2023) or active learning methods (Mittal et al., 2023) does not guarantee consistency or coherent annotations. Alternatively, annotating 3D parts of already existing 3D CAD models emerges as a cheaper and more scalable solution. Since all parts are discernible within the 3D space, the inherent ambiguity associated with part annotations in 2D images because of self-occlusion is mitigated, facilitating more precise and detailed part definitions. By rendering these 3D CAD models with 3D part annotations, it becomes feasible to generate unlimited synthetic data samples with precise control over 3D viewpoints (and individual parts), thereby producing highly accurate part segmentation masks.

Despite these advantages, existing datasets created by simply rendering 3D CAD models (Mu et al., 2020; Liu et al., 2022; Peng et al., 2024) severely lack realism, presumably due to the limited availability of high-quality textures in the CAD models. Additionally, rendered objects typically exhibit unrealistic interactions with their backgrounds, as the rendering process lacks constraints to ensure accurate context integration. This often enlarges the domain gap between the synthetic data and real-world data, reducing its utility. A recent work (Ma et al., 2024) has tried to ameliorate the aforementioned problems by using latent diffusion models with text prompts, coupled with ControlNet (Zhang et al., 2023b), to generate more realistic synthetic images conditioned on edge maps of rendered images. However, this approach is limited by its focus on merely coarse 3D object pose outputs (*i.e.* object viewpoints) on rigid objects. It falls short in generating high-quality samples that are consistent with the visual prompts for nonrigid objects, such as animals, which exhibit a wide range of poses and shapes.

In this work, we introduce Diffusion-generated Synthetic Parts (DSPart), a comprehensive part dataset comprising annotated 3D CAD models and realistic 2D images rendered with 3D-DST (Ma et al., 2024), which serves as a nontrivial extension to 3D-DST in the part context. Specifically, for rigid object categories, we present DSPart-Rigid, which includes 475 3D CAD models from 50 rigid object

| dataset | City.-PP (Meletis et al., 2020) | Pascal-Part (Chen et al., 2014) | ADE20k (Zhou et al., 2017) | PartImageNet (He et al., 2022) | PACO (Ramanathan et al., 2023) | UDAPart (Liu et al., 2022) | CC-SSL (Mu et al., 2020) | 3DCOMPAT++ (Slim et al., 2023) | DSPart (ours) |
|---|---|---|---|---|---|---|---|---|---|
| image domain | real | real | real | real | real | synthetic | synthetic | synthetic | synthetic |
| #images | 3.5k | 19k | 12.6k | 24k | 76.7k | 126k | 20k | 160M | 100k |
| #categories | 5 | 20 | 80 | 158 | 75 | 5 | 2 | 41 | 90 |
| #non-rigid categories | 1 | 8 | 1 | 122 | 1 | 0 | 2 | 0 | 40 |
| #avg parts per object | 4.60 | 9.65 | 7.08 | 3.85 | 6.08 | 24 | 7 | 9.98 | 6.93 |
| #3D parts | - | - | - | - | - | 120 | 14 | 100k | 24.5k |
| #3D instances | - | - | - | - | - | 21 | 2 | 10k | 3540 |
| real image context | ✓ | ✓ | ✓ | ✓ | ✓ | ✗ | ✗ | ✗ | ✓* |
| auto-scalability** | ✗ | ✗ | ✗ | ✗ | ✗ | ✓ | ✓ | ✓ | ✓ |

Table 1: **Statistical comparison between DSPart and other publicly available image part datasets.** DSPart provides dense part annotations (6.93 per image) over a wide range of categories (90), including the most challenging non-rigid animal categories (40). It offers a substantial number of rendered images (100k), along with annotated 3D parts (24.5k), and can be scaled up further if needed. Compared to the recent 3DCOMPAT++ (Slim et al., 2023), DSPart benefits from real image context synthesized by diffusion models. *: realistic synthesized context. **: the ability to generate more images with part masks without additional annotations.

categories annotated with fine-grained part definitions. Using the 3D-DST synthesis pipeline, we generate 100 images per 3D CAD model, resulting in a total of 48K synthetic images. For nonrigid object categories, we introduce DSPart-Animal, which has 52K synthetic animal images. We use 3,065 SMAL (Zuffi et al., 2017) models fitted poses from 40 animal categories, which are obtained from the training data of Animal3D Dataset (Xu et al., 2023). Since all SMAL models share the same vertex IDs, we only need one 3D part annotation for all the poses. To solve the high failure rate of generated animal images which the proposed filtering strategy in 3D-DST can not effectively mitigate, we propose a more suitable filter, namely PRF. PRF uses metrics and algorithms from 3D animal pose estimation that focuses not only on 3D rigid body pose but also the accuracy of articulations. It has been proven to be more efficient and effective in filtering out low-quality synthetic animal samples, which makes scaling the dataset easier.

Subsequently, we perform 2D part segmentation on both DSPart-Rigid and DSPart-Animal in both synthetic-only and synthetic-to-real scenarios to evaluate their performance on PartImageNet (He et al., 2022). We also conduct experiments on PascalPart (Chen et al., 2014) for fine-grained scenarios.

The experimental results validate the effectiveness of the proposed DSPart as synthetic training data with less domain gap and better generalizability compared to previous synthetic part datasets, attributed to its accurate 3D annotated CAD models and high-quality 2D rendered images. In summary, our contributions in this work include the following:

- We present DSPart, comprising DSPart-Rigid and DSPart-Animal, a large-scale part dataset featuring 475 rigid CAD models and 3,065 fitted animal poses in 3D, along with 48K synthetic images of rigid objects and 52K synthetic images of animals. DSPart significantly surpasses existing part datasets in both real and synthetic domains in terms of 3D part annotations and realistic image context, as shown in Table 1.

- We introduce PRF, an efficient and effective filter mechanism designed to exclude low-quality animal synthetic images due to the large variation in pose and shape, thus serving as a robust extension of the 3D-DST synthesis pipeline.

- We conduct extensive experiments on DSPart for 2D part segmentation to demonstrate that DSPart significantly outperforms existing synthetic part datasets in terms of data quality as synthetic training data.

## 2 RELATED WORKS

**Training with synthetic data.** Synthetic data has gained significant attention in generating labeled data to train computer vision models that require extensive annotations (Rombach et al., 2022; Zhang et al., 2023a). The "training with synthetic data" methods can be categorized into two groups: 1) *2D-based methods* employ recent generative models like GANs and diffusion models to create photo-realistic images for model training, pre-training, or data augmentation (Baranchuk et al., 2022; Liu et al., 2021; Dosovitskiy et al., 2015; Sun et al., 2021). However, while offering image realism, they often do not incorporate explicit 3D structural data, which limits their utility in tasks requiring

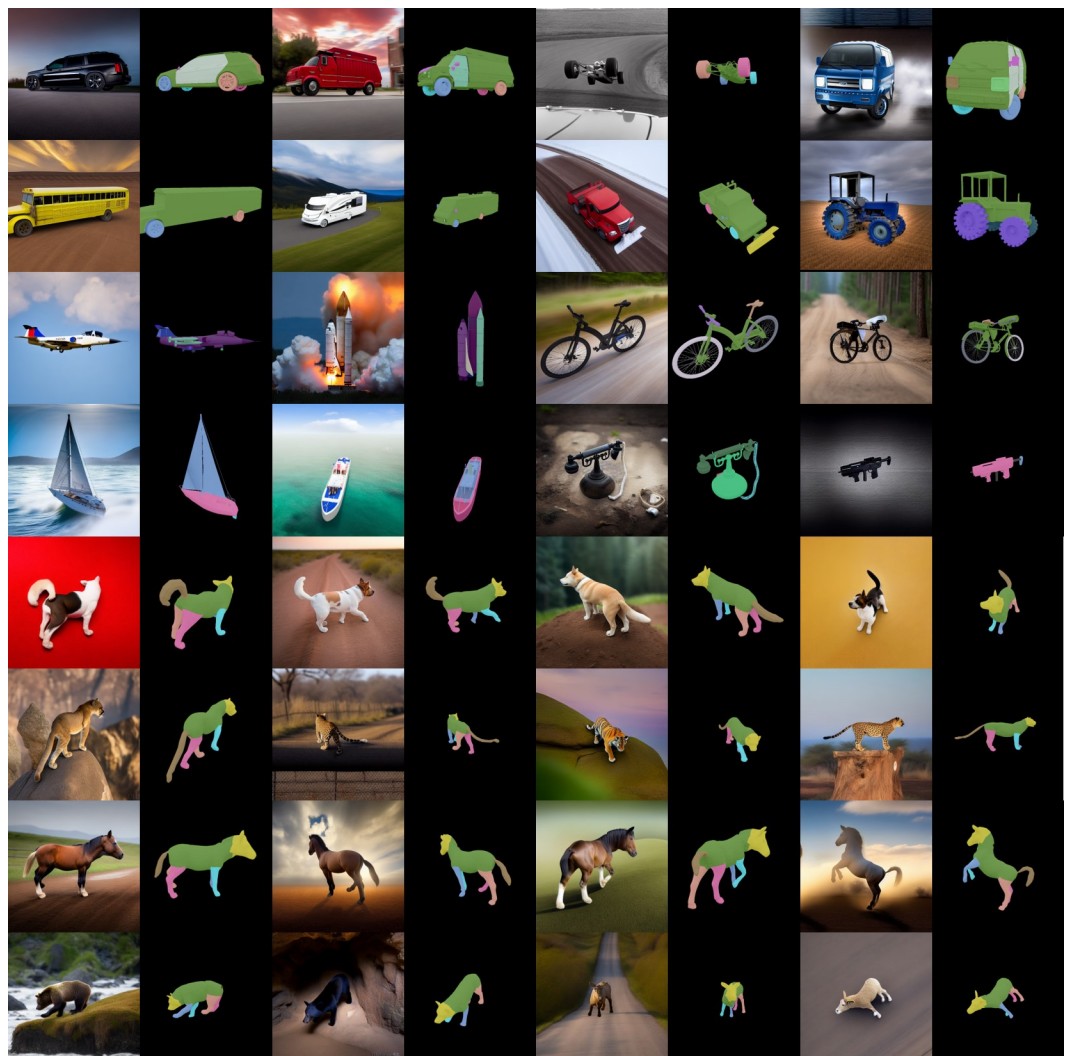

**Figure 2: Qualitative examples of DSPart dataset.** Examples are generated by 16 3D CAD models from 16 different rigid object categories and 16 different SMAL poses from 12 animal species. The examples exhibit realistic textures and image contexts under varying viewpoints and shapes.

precise 3D annotations. 2) *3D-based methods* involve using advanced rendering technologies and physical simulations to create environments and objects with realistic textures and physics (Greff et al., 2022; Zheng et al., 2020). Although these methods provide valuable 3D insights, their diversity is generally restricted by the texture variability of the 3D models used. In our work, we use the DST (Wu et al., 2023), which integrates 3D control directly into the diffusion process. This allows for the synthesis of images that not only adhere to photo-realistic standards but also encapsulate accurate 3D annotations.

**Learning part models from synthetic data.** Learning part models from synthetic data has garnered increasing attention, primarily due to the difficulty of obtaining large-scale real part-level annotations. Mu et al. (2020) conduct an early exploration in a multi-task learning setting, where part segmentation and object pose estimation are jointly learned. They show that this multi-task learning could enhance object pose estimation, but they do not fully explore the potential of part segmentation itself. Liu et al. (2022) propose a cross-domain geometric matching module to align predictions and warp synthetic results to the real domain, improving alignment and performance. Peng et al. (2024) leverage recent syn-to-real domain transfer algorithms, augmenting them with a class-balanced pseudo-label re-weighting mechanism to better align synthetic results with the real domain. In this work, we focus on improving the quality of synthetic data itself rather than proposing new methods for syn-to-real

transfer. Consequently, we present DSPart and demonstrate that re-training existing algorithms using DSPart can significantly enhance performance in 2D part segmentation tasks.

## 3 DATASET CONSTRUCTION

In this section, we introduce how we construct our DSPart synthetic dataset. The dataset consists of two parts, DSPart-Rigid (Section 3.1) and DSPart-Animal (Section 3.2). The statistical comparison between DSPart and other publicly available image part datasets are presented in Table 1. The example of images and annotated parts are shown in Figure 2.

### 3.1 DSPART-RIGID

DSPart-Rigid aims to provide fine-grained 2D part masks and 3D part annotations for common rigid models in the real world. Existing 3D part datasets (Mo et al., 2019; Slim et al., 2023) have annotated a large amount of 3D shapes, but the object categories are limited. PartNet (Mo et al., 2019) have annotated 24 categories of indoor rigid objects, which makes it hard to find real image part datasets for evaluation since existing ones focus on common outdoor rigid objects such as cars (Meletis et al., 2020; Chen et al., 2014; Zhou et al., 2017; He et al., 2022). 3DCOMPAT++ (Slim et al., 2023) have annotated 41 object categories, which include vehicles, but these categories have much fewer 3D CAD models compared to other indoor objects (*e.g.* only 36 CAD models for the car category) and are not diverse enough in shape. Furthermore, it does not further classify vehicle categories into more fine-grained ImageNet1k (Deng et al., 2009) classes, which increases the possibility of failure cases in the DST-3D (Ma et al., 2024) pipeline because of the missing class-specific details of prompts. For instance, for a minivan CAD model, if the text prompt describes it as a general car, the generated synthetic images are more likely to be inconsistent with the visual conditions.

To make our synthetic dataset capable of evaluation on existing real part datasets and also avoid the aforementioned class ambiguity issue, we carefully select 50 rigid object categories that are consistent with ImageNet1k class definitions. We manually classify some 3D shapes from ShapeNet (Chang et al., 2015) and Objaverse (Deitke et al., 2022) to these fine-grained categories. Subsequently, we manage to select around 10 CAD models with large shape differences for each category since we want every CAD model annotated to be representative. We recruit 25 annotators for data annotation and more details are provided in Appendix A.3.

**Annotation scheme. (i) What parts to annotate per category:** One of the key challenges in annotating parts of 3D CAD models is the ambiguity of object part selection (*e.g.* how to annotate the parts of a space shuttle). We divide our 50 rigid-object categories into five super-categories: car, airplane, bicycle, boat, and tool. We classify the super-category of each object category based on Wikidata and common knowledge. We then analyze what object parts are important in cognition and are tractable to be annotated in real images. Subsequently, we create part definition templates for each super-category, except tools, as the shapes of tools vary significantly. Therefore, we define part definitions individually for each tool category. Note that our part definitions are all recognizable from the object surface and we do not define parts that are internal structures. Annotators will check the shapes of their assigned CAD models first and then revise the provided part definitions only if necessary. Please refer to Appendix A.1 for the part taxonomy. **(ii) What principles to select part vertices:** We design several principles in selecting part vertices to guide the annotators to ensure high-quality and consistent 3D part annotations. Firstly, the annotated part vertex groups should be disjoint sets, and the union of all groups should contain every vertex in the original CAD models. Secondly, if a mesh face belongs to two connected parts, the annotator should not assign all three vertices to one part and should still assign the vertices based on where they are located. **(iii) Annotation quality inspection:** We have 2 iterations of the annotation inspection, including correctness check and revision. The first iteration of annotation inspection is done by selected annotators whose annotations are high-quality during the annotation process, and the annotators will inspect the object categories that belong to the same super-categories of what they annotated in the annotation process. Then, we perform the second iteration of annotation inspection.

### 3.2 DSPART-ANIMAL

Existing synthetic animal datasets (Mu et al., 2020; Jiang & Ostadabbas, 2023) have less than ten animal species due to the limited availability of 3D animal CAD models. DSPart-Animal

aims to achieve a larger scale in animal species to train robust models for the in-the-wild animals. SMAL (Zuffi et al., 2017) models can fit various quadruped animals based on their shape priors. However, SMAL models have no textures, which is the crucial reason that prevents people from using them to generate synthetic images. Existing works (Zuffi et al., 2018; 2019) predict the textures using real images and paste them on the SMAL models, but these texture prediction methods do not generalize well enough beyond the distribution of textures observed during training. By leveraging the powerful latent diffusion model in the 3D-DST pipeline (Ma et al., 2024), DSPart-Animal can have realistic and diverse textures that generalize better.

**Part-attention Regressor Filter (PRF).** Unfortunately, the 3D-DST (Ma et al., 2024) images of animals exhibit low quality due to the wide variation in pose and shape. The 3D-DST proposes the K-fold consistency filter (KCF) to filter out noisy samples with inaccurate 3D annotations. It creates the K-fold splits of synthetic samples and for each train-val split, it trains a state-of-the-art 3D pose estimation model (Ma et al., 2022) and evaluates it on the validation split. Each sample in the validation split will be classified as noisy, with potentially inaccurate 3D annotations, if the confidence score associated with the 3D annotation falls below a specified threshold.

However, KCF is trained on unfiltered images, which are noisy in the case of animals, leading to unreliable model predictions. Furthermore, KCF relies on the pose error given by the angle between the predicted rotation matrix and the ground truth rotation matrix (Zhou et al., 2018), which is not an effective filtering metric for animals due to their high degree of articulation and deformation.

We then look at the metrics used for 3D human pose estimation and 3D animal pose estimation models trained from reliable data. We recruit three annotators to do the human filtering on around 46k synthetic animal images (*i.e.* 15 viewpoints for each SMAL pose). The filtering is based on three principles: (1) the consistency of leg positions and leg length; (2) the consistency of head orientation and neck deformation; (3) the face of the animal is recognizable. It takes around 100 working hours, and we finally get 13k synthetic animal images with reasonably accurate 3D annotations. We train the 3D animal pose estimation algorithms from (Xu et al., 2023) with these 13k images and PARE (Kocabas et al., 2021a) exhibit the most robust cross-domain performance. The results and justification are provided in Appendix B.

We employ two PARE models for filtering: a model trained exclusively on our human-filtered synthetic data for $1,000$ epochs, and a model pre-trained on our synthetic data for 100 epochs, followed by $1,000$ epochs of fine-tuning on the Animal3D training set. We specify thresholds for three metrics: Procrustes-aligned mean per joint position error (PA-MPJPE), scale-aligned mean per joint position error (S-MPJPE), and 2D Percentage of Correct Keypoints (PCK). We choose the intersection of filtered image sets from the aforementioned two PARE models as the model-filtered samples. We demonstrate the quality of model-filtered samples in Section 4.3, which guarantees the scalability of DSPart-Animal. We name the filtering process as Part-attention Regressor Filter (PRF) inspired by the name of PARE, and we use it to filter the left 39k images in DSPart-Animal.

## 4 EXPERIMENTS

To validate the effectiveness of DSPart dataset, we conduct evaluations on 2D object part segmentation Section 4.2. In Section 4.3, we exhibit the effectiveness and efficiency of PRF and thus demonstrate the scalability of DSPart-Animal. Furthermore, we demonstrate the benefits of diffusion-generated textures and contexts (*i.e.* background) in Section 4.4.

### 4.1 EXPERIMENTS SETUP

**Datasets.** We evaluate the 2D object part segmentation mainly on PartImageNet (He et al., 2022), as its category and class mapping aligns most closely with DSPart among existing real part datasets. Specifically, we evaluate the performance of DSPart on its test set of 4 super-categories of rigid objects and the quadruped category. For each super-category, we exclusively use synthetic images from that super-category for training. Note that the training data we use for animal experiments is the 13k human-filtered animal images if not specified. To access the performance of DSPart-Rigid, we compare it against UDAPart (Liu et al., 2022), a specialized dataset for vehicle part segmentation. The only existing synthetic animal dataset with 2D part annotations is CC-SSL (Mu et al., 2020), which sourced its animal CAD models from the Unreal Engine Marketplace. CC-SSL includes 10k horse images and 10k tiger images. We use this dataset as a benchmark to assess the performance of

| datasets | supervision | architecture | car | airplane | boat | bicycle |
|---|---|---|---|---|---|---|
| UDAPart | Syn-only | SegFormer | 58.55 | 39.10 | - | 35.79 |
| | UDA | DAFormer | 65.59 | 47.73 | - | 39.67 |
| DSPart-Rigid | Syn-only | SegFormer | 58.22 | 58.04 | 61.63 | 37.03 |
| | UDA | DAFormer | 72.67 | 59.45 | 65.31 | 51.55 |
| PartImageNet | Real-only | SegFormer | 74.08 | 69.98 | 81.71 | 66.27 |
| DSPart-Rigid + PartImageNet | Syn+Real | SegFormer* | **75.30** | **72.43** | **82.51** | **66.70** |

Table 2: **Part segmentation results on 4 rigid super-categories**. The average mIoU of parts for each super-category is reported. *: models trained with synthetic data and real data successively for each iteration. Numbers are averaged over 3 random seeds.

DSPart-Animal. Additionally, we provide evaluation results of horse category on PascalPart (Chen et al., 2014) to access the performance for more fine-grained part definitions and more challenging scenarios (*e.g.* severe occlusion, truncation, multiple objects).

**Training setup.** We conduct our part segmentation experiments using SegFormer (Xie et al., 2021) and DAFormer (Hoyer et al., 2022) to serve as the architectures for fully supervised settings and unsupervised domain adaptation (UDA) respectively. MiT-b5 (Xie et al., 2021) pre-trained on ImageNet-1k is adopted as the backbone. The real training data for the UDA settings are training images of each super-category from the train set of PartImageNet. Due to varying image numbers across different synthetic datasets, we use iterations rather than epochs for a fairer comparison of evaluation results. Unless specified, synthetic-only models are trained with a batch size of 2 on a single GPU for 10k iterations to avoid overfitting in the synthetic domain, while all other settings are trained for 30k iterations. Please refer to Appendix C.1 for more implementation details.

### 4.2 2D PART SEGMENTATION

**DSPart-Rigid.** Table 2 summarizes our results of SegFormer (Xie et al., 2021) and DAFormer (Hoyer et al., 2022) on the PartImageNet test set when training with different datasets under various settings.

Specifically, although training with DSPart-Rigid performs similarly to training with UDAPart (Liu et al., 2022) under the Syn-only setting, it significantly improves performance in the unsupervised domain adaptation setting, where knowledge learned from the synthetic dataset is better transferred to the real domain. Concretely, when training with DSPart-Rigid, DAFormer achieves improvements of 7.08, 11.72, and 11.88 mIoU on car, airplane, and bicycle categories, respectively. Remarkably, it even achieves performance comparable to training with real images on certain relatively easy categories (*i.e.*, 72.67 *vs.* 74.08 mIoU on the car category). We hypothesize that the main reason behind this improvement is that better cross-domain features are learned from DSPart-Rigid because of the diverse and realistic textures and image contexts.

The reason for no significant improvements in the Syn-only setting is that DSPart-Rigid still has a small fraction of noisy samples (*i.e.* the image content is not consistent with the 3D annotation) even after using K-fold consistency filter (KCF) (Ma et al., 2024). This issue is mitigated in the UDA settings since the pseudo-labels are generated in online self-training that uses exponentially moving averages to update. It increases the stability of the pseudo-label predictions, and thus, the quality of pseudo-labels will not be influenced significantly by minor noisy samples.

Furthermore, we observe that combining real domain training data with DSPart-Rigid and training SegFormer in a sequential manner—where synthetic and real data are fed into the model iteratively—yields further performance improvements. This showcases the potential of utilizing synthetic data to boost state-of-the-art performance.

**DSPart-Animal.** Table 3 and Table 4 present the results of SegFormer and DAFormer on the animal part segmentation benchmark when trained using different data sources under various supervision settings. Notably, DSPart-Animal demonstrates a robust capability to enhance model training under both Syn-only and UDA settings, owing to its more diverse and realistic images. In Table 4, an interesting observation is that although CC-SSL synthetic data performs well in synthetic-only settings

| datasets | supervision | architecture | head | torso | leg | tail | background | mIoU |
|---|---|---|---|---|---|---|---|---|
| CC-SSL | Syn-only | SegFormer | 23.84 | 34.64 | 30.38 | 9.53 | 78.92 | 35.46 |
| | UDA | DAFormer | 41.56 | 45.14 | 44.66 | 18.92 | 89.56 | 47.97 |
| DSPart-Animal | Syn-only | SegFormer | 48.94 | 39.53 | 32.09 | 16.40 | 85.43 | 44.48 |
| | UDA | DAFormer | 52.63 | 52.27 | 48.67 | 25.00 | 94.55 | 54.62 |
| PartImageNet | Real-only | SegFormer | 85.91 | 72.42 | 60.51 | 50.23 | 96.71 | 73.16 |
| DSPart-Animal + PartImageNet | Syn+Real | SegFormer* | **87.43** | **74.19** | **63.93** | **55.31** | **96.75** | **75.52** |

**Table 3: Part segmentation results on animals**. DSPart-Animal outperforms CC-SSL in all settings. *: models trained with synthetic data and real data successively for each iteration. Numbers are averaged over 3 seeds.

| datasets | supervision | architecture | head | fl-leg | fr-leg | bl-leg | br-leg | tail | torso | bg | mIoU |
|---|---|---|---|---|---|---|---|---|---|---|---|
| CC-SSL | Syn-only | SegFormer | 33.29 | 16.13 | 13.61 | 11.22 | 17.32 | 33.49 | 45.65 | 85.35 | 32.01 |
| | UDA | DAFormer | 34.47 | 0.02 | 20.44 | 0.01 | 0.03 | 40.10 | 59.60 | 93.65 | 31.04 |
| DSPart-Animal | Syn-only | SegFormer | 48.65 | 15.91 | 15.76 | 20.74 | 10.72 | 22.90 | 45.85 | 88.21 | 33.59 |
| | UDA | DAFormer | 59.88 | 0.07 | **27.81** | 0.11 | **24.20** | 53.81 | 57.92 | 94.21 | 39.75 |
| PascalPart | Real-only | SegFormer | 75.57 | **17.55** | 13.61 | 15.86 | 17.29 | 54.33 | 77.50 | 95.89 | 45.95 |
| DSPart-Animal + PascalPart | Syn+Real | SegFormer* | **85.08** | 16.25 | 25.19 | **24.26** | 21.93 | **55.64** | 83.46 | 95.95 | **50.97** |

**Table 4: Part segmentation results of horse category on Pascal-Part**. Only horse images are used for training. DSPart-Animal outperforms CC-SSL in all settings. *: models trained with synthetic data and real data successively for each iteration. Numbers are averaged over 3 random seeds. 'fl-leg', 'fr-leg', 'bl-leg', 'br-leg', and 'bg' stand for front-left-leg, front-right-leg, back-left-leg, back-right-leg, and background respectively.

and is comparable to DSPart-Animal, its UDA performance is even worse than in the synthetic-only scenarios. We hypothesize that DAFormer trained on CC-SSL may produce very low-quality pseudo-labels during the early stages of training and have more severe spatial ambiguity issues (*i.e.* very low IoU for some legs). Furthermore, the DAFormer performance of DSPart-Animal is close to the real-only results with even extreme low IoU of front-left and front-right legs. There is a promising potential for future UDA methods to solve these spatial ambiguity issues and achieve comparable results to supervised training on real.

### 4.3  ABLATION ON THE EFFECTIVENESS OF PRF

Table 5 shows the ablation studies on different filtering mechanisms. Specifically, we compare four types of filtering: no filter, KCF filter (Ma et al., 2024), and the proposed PRF filter. Additionally, we provide a potential upper bound with our human-filtered images. Except for the human-filtered data obtained from 46k images, all other filtering strategies are applied to a separate set of 150k images. From the table, we observe that the KCF filter proposed in 3D-DST (Ma et al., 2024) achieves performance similar to unfiltered results. This suggests that the KCF filter may not be effective in the context of animals since the pose error calculated by the rotation matrix is not enough to measure the accuracy of articulations and deformation.

The proposed PRF filter significantly outperforms prior methods, achieving a mIoU improvement of 9.45 and 6.25 over unfiltered and KCF-filtered data, respectively. Furthermore, the PRF-filtered result under the UDA setting is close to the human-filtered result, which demonstrates that we can maintain the scalability of 3D-DST (Ma et al., 2024) with PRF filter when extending to animals. Moreover, the proposed PRF filter is more cost-effective to apply. The KCF filter (Ma et al., 2024) necessitates training multiple models (*e.g.* 5) on different subsets of the generated data and then performing cross-validation experiments to filter out low-quality data, which takes around 50 hours to train all the models. The human filter approach requires approximately 100 hours of manual labor to identify and exclude undesirable images. In contrast, the PRF filter only needs a pre-trained model and can directly run inference on the generated images at scale without the need for retraining. This process takes only about 20 minutes, making it significantly more efficient.

### 4.4  ABLATION ON DIFFUSION-GENERATED TEXTURES AND BACKGROUND

Table 6 exhibits the ablation studies on diffusion-generated object textures and realistic synthesized context (*i.e.* background). We perform the comparison on the airplane category of 3DCoM-

| filter | cost | supervision | architecture | head | torso | leg | tail | background | mIoU |
|---|---|---|---|---|---|---|---|---|---|
| - | None | Syn-only | SegFormer | 18.28 | 21.51 | 10.48 | 5.71 | 74.90 | 26.17 |
| | | UDA | DAFormer | 33.39 | 39.77 | 45.96 | 12.71 | 87.80 | 43.93 |
| K-fold Consistency (KCF) | 50 training hours | Syn-only | SegFormer | 20.67 | 20.80 | 16.54 | 9.91 | 75.80 | 28.74 |
| | | UDA | DAFormer | 31.59 | 43.46 | 48.83 | 20.19 | 89.46 | 46.71 |
| Part-attention Regressor (PRF) | 20 mins for inference | Syn-only | SegFormer | 43.33 | 28.82 | 25.56 | 13.12 | 80.09 | 38.19 |
| | | UDA | DAFormer | **52.85** | 48.57 | 44.64 | **31.30** | 87.45 | 52.96 |
| Human | 100 hours human working | Syn-only | SegFormer | 48.94 | 39.53 | 32.09 | 16.40 | 85.43 | 44.48 |
| | | UDA | DAFormer | 52.63 | **52.27** | **48.67** | 25.00 | **94.55** | **54.62** |

**Table 5: Comparison of filtering strategies for DSPart-Animal dataset**. The human-filtered setting has the best performance, while PRF has the best performance-time trade-off. Numbers are averaged over 3 seeds.

| datasets | supervision | architecture | body | wing | engine | tail | bg | mIoU |
|---|---|---|---|---|---|---|---|---|
| 3DCoMPaT++ (white background) | Syn-only | SegFormer | 25.45 | 11.28 | 7.64 | 5.64 | 83.43 | 26.69 |
| | UDA | DAFormer | 45.63 | 24.32 | 5.18 | 3.42 | 85.40 | 32.79 |
| 3DCoMPaT++ foreground w/ diffusion background | Syn-only | SegFormer | 24.89 | 20.59 | 9.54 | 5.90 | 87.11 | 29.61 |
| | UDA | DAFormer | 44.76 | 25.62 | 6.79 | 3.81 | 92.33 | 34.66 |
| Diffusion generated (Ours) | Syn-only | SegFormer | 33.01 | 19.63 | **13.00** | **15.68** | 90.20 | 34.30 |
| | UDA | DAFormer | **56.43** | **26.37** | 10.71 | 9.61 | **94.13** | **39.45** |

**Table 6: Ablation on diffusion-generated textures and background**. "Diffusion generated" refers to synthetic data generated following our pipeline that uses the same 32 CAD models and viewpoints with 3DCoMPaT++. "3DCoMPaT++ foreground w/ diffusion background" is generated by replacing the foreground object of diffusion-generated data with the foreground from 3DCoMPaT++. The results are evaluated on the PartImageNet test set of airplanes. Numbers are averaged over 3 random seeds.

PaT++ (Slim et al., 2023). Synthetic images in 3DCoMPaT++ have more diverse textures by compositing a wide range of materials on different parts, which is a good representative of direct rendering methods. Additionally, their synthetic images are rendered with a uniformly white background which is beneficial for evaluating the effectiveness of the "realistic synthesized context" we proposed in table 1. "3DCoMPaT++ foreground w/ diffusion background" data is generated by replacing the foreground object of our diffusion-generated data with the foreground from 3DCoMPaT++. The evaluation is performed on the PartImageNet test set of airplanes.

In Table 6, the diffusion-generated data following our pipeline uses the same 32 CAD models and viewpoints with 3DCoMPaT++. As observed in the table, diffusion-generated object textures lead to better performance (*i.e.* more realistic) than materials provided in 3DCoMPaT++ with improvements of 4.61 mIoU under the Syn-only setting and 4.79 mIoU under the UDA setting. The "realistic synthesized context" also benefits by comparing it to the white background, which is significant for observing that the background IoU improves from 85.40 to 92.33 under the UDA setting. Furthermore, the current diffusion-generated data leads to worse performance on PartImageNet compared to airplane results in Table 2. We hypothesize two possible reasons: 1) the shape space we selected in DSPart-Rigid is bigger since we selected CAD models with large shape differences for each category and tried to make every CAD model we annotated representative; 2) 3DCoMPaT++ only has 8 viewpoints for each 3D shape while we have 100 viewpoints in DSPart-Rigid.

## 5 CONCLUSIONS

We present DSPart, comprising DSPart-Rigid and DSPart-Animal, a large-scale part dataset featuring 475 rigid CAD models and 3,065 fitted animal poses in 3D, along with 48K synthetic images of rigid objects and 52K synthetic images of animals. DSPart significantly surpasses existing part datasets in both real and synthetic domains in terms of 3D part annotations and realistic synthesized image context. We introduce PRF, an efficient and effective filter mechanism designed to exclude low-quality animal synthetic images due to the large variation in pose and shape, thus serving as a robust extension of the 3D-DST synthesis pipeline and maintaining scalability for animals. We conduct extensive experiments on DSPart for 2D part segmentation to demonstrate that DSPart vastly outperforms existing synthetic part datasets in data quality when used as synthetic training data.

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

| class ID | class name | parts taxonomy |
|---|---|---|
| **DSPart-Rigid** | | |
| n02690373 | airliner | engine†, fuselage, wing†, vertical_stabilizer, wheel(front, back_left, back_right), horizontal_stabilizer† |
| n02701002 | ambulance | wheel(front_left, front_right, back_left, back_right), door†, front_trunk, back_trunk, head_light†, frame, rearview† |
| n02749479 | gun | buttstock,magazine,barrel, gunbody |
| n02804414 | bassinet | stand, frame |
| n02814533 | beach wagon | wheel(front_left, front_right, back_left, back_right), door†, front_trunk, back_trunk, head_light†, frame, rearview† |
| n02835271 | bicycle built for two | wheels(back, front), frame(paddle), handlebar, saddle |
| n02906734 | broom | handle, head |
| n02981792 | catamaran | sail, body |
| n03063689 | coffeepot | spout, body, handle |
| n03100240 | convertible | wheel(front_left, front_right, back_left, back_right), door†, front_trunk, back_trunk, head_light†, frame, rearview† |
| n03187595 | dial telephone | handset, dial, host, cord |
| n03272562 | electric locomotive | wheel†, door†, frame, rearview† |
| n03344393 | fireboat | top, body |
| n03345487 | fire engine | wheel†, door†, ladder_and_pump, frame, rearview† |
| n03417042 | garbage truck | wheel†, frame, front trunk, garbage_container |
| n03444034 | go-kart | wheel(front_left, front_right, back_left, back_right), frame, seat, engine |
| n03445924 | golfcart | wheel, frame, seat |
| n03481172 | hammer | handle, head |
| n03496892 | harvester | wheel†, frame, cutter, mirror |
| n03498962 | hatchet | handle, head |
| n03594945 | jeep | wheel(front_left, front_right, back_left, back_right), door†, frame, front_trunk, back_trunk, rearview† |
| n03599486 | jinrikisha | wheel(front, back_left, back_right), saddle, frame |
| n03642806 | laptop | keyboard, screen, body, touchpad |
| n03649909 | mower | wheel(front_left, front_right, back_left, back_right), steering_wheel, shaft, frame |
| n03670208 | limo | wheel(front_left, front_right, back_left, back_right), frame, rearview†, door†, head_light† |
| n03673027 | ocean liner | top, body |
| n03769881 | minibus | wheel(front_left, front_right, back_left, back_right), frame, door, rearview† |
| n03770679 | minivan | wheel(front_left, front_right, back_left, back_right), frame, door†, head_light† |
| n03785016 | moped | wheel(front, back), handlebar, frame, rearview |
| n03792782 | mountain bike | wheels(back, front), frame, handlebar, saddle |
| n03891251 | park bench | arm, backrest, beam, seat, leg |
| n03947888 | pirate ship | sail, body |
| n03977966 | Police car | wheel(front_left, front_right, back_left, back_right), door†, front_trunk, back_trunk, frame, rearview† |
| n04037443 | race car | wheel(front_left, front_right, back_left, back_right), door†, front_trunk, back_trunk, head_light†, frame, rearview† |
| n04065272 | recreational vehicle | wheel(front_left, front_right, back_left, back_right), door†, front_trunk, back_trunk, head_light†, frame, rearview† |
| n04146614 | school bus | wheel(front_left, front_right, back_left, back_right), frame, head_light†, door†, rearview† |
| n04147183 | schooner | sail, bottom |
| n04204347 | shopping cart | wheel(front_left, front_right, back_left, back_right), basket, handle, frame |
| n04252225 | snowplow | wheel(front_left, front_right, back_left, back_right), frame, rearview†, cutter |
| n04266014 | space shuttle | engine†, fuselage, wing†, vertical_stabilizer, wheel†, horizontal_stabilizer† |
| n04285008 | sports car | wheel(front_left, front_right, back_left, back_right), door†, front_trunk, back_trunk, head_light†, frame, rearview† |
| n04465501 | tractor | wheel†, door†, arm_and_loader, frame, rearview† |
| n04467665 | trailer truck | wheel†, door†, front_trunk, trailer, head_light†, frame, rearview† |
| n04482393 | tricycle, trike, velocipede | wheels†, frame, handlebar, saddle, cargo_box |
| n04483307 | trimaran | sail, body |
| n04487081 | trolleybus, trolley coach, trackless trolley | wheel(front_left, front_right, back_left, back_right), frame, door†, head_light† |
| n04507155 | umbrella | handle, canopy, frame |
| n04509417 | unicycle, monocycle | wheels, frame, saddle |
| n04552348 | warplane, military plane | engine†, fuselage, wing†, vertical_stabilizer, wheel(middle, left, right), horizontal_stabilizer† |
| n04612504 | yawl | sail, body |
| **DSPart-Animal** | | |
| n02085782 | japanese spaniel | head, torso, front_left_leg, front_right_leg, back_left_leg, back_right_leg, and tail |
| n02089867 | walker hound | head, torso, front_left_leg, front_right_leg, back_left_leg, back_right_leg, and tail |
| n02090379 | redbone | head, torso, front_left_leg, front_right_leg, back_left_leg, back_right_leg, and tail |
| n02091831 | saluki | head, torso, front_left_leg, front_right_leg, back_left_leg, back_right_leg, and tail |
| n02092339 | weimaraner | head, torso, front_left_leg, front_right_leg, back_left_leg, back_right_leg, and tail |
| n02096177 | cairn | head, torso, front_left_leg, front_right_leg, back_left_leg, back_right_leg, and tail |
| n02096585 | boston bull | head, torso, front_left_leg, front_right_leg, back_left_leg, back_right_leg, and tail |
| n02097474 | tibetan terrier | head, torso, front_left_leg, front_right_leg, back_left_leg, back_right_leg, and tail |
| n02098105 | soft-coated wheaten terrier | head, torso, front_left_leg, front_right_leg, back_left_leg, back_right_leg, and tail |
| n02099601 | golden retriever | head, torso, front_left_leg, front_right_leg, back_left_leg, back_right_leg, and tail |
| n02100583 | vizsla | head, torso, front_left_leg, front_right_leg, back_left_leg, back_right_leg, and tail |
| n02101006 | Gordon setter | head, torso, front_left_leg, front_right_leg, back_left_leg, back_right_leg, and tail |
| n02101388 | brittany spaniel | head, torso, front_left_leg, front_right_leg, back_left_leg, back_right_leg, and tail |
| n02102040 | english springer | head, torso, front_left_leg, front_right_leg, back_left_leg, back_right_leg, and tail |
| n02102973 | irish water spaniel | head, torso, front_left_leg, front_right_leg, back_left_leg, back_right_leg, and tail |
| n02109525 | saint Bernard | head, torso, front_left_leg, front_right_leg, back_left_leg, back_right_leg, and tail |
| n02109961 | eskimo dog | head, torso, front_left_leg, front_right_leg, back_left_leg, back_right_leg, and tail |
| n02112137 | chow | head, torso, front_left_leg, front_right_leg, back_left_leg, back_right_leg, and tail |
| n02114367 | timber wolf | head, torso, front_left_leg, front_right_leg, back_left_leg, back_right_leg, and tail |
| n02120079 | arctic fox | head, torso, front_left_leg, front_right_leg, back_left_leg, back_right_leg, and tail |
| n02124075 | egyptian cat | head, torso, front_left_leg, front_right_leg, back_left_leg, back_right_leg, and tail |
| n02125311 | cougar | head, torso, front_left_leg, front_right_leg, back_left_leg, back_right_leg, and tail |
| n02128385 | leopard | head, torso, front_left_leg, front_right_leg, back_left_leg, back_right_leg, and tail |
| n02129604 | tiger | head, torso, front_left_leg, front_right_leg, back_left_leg, back_right_leg, and tail |
| n02130308 | cheetah | head, torso, front_left_leg, front_right_leg, back_left_leg, back_right_leg, and tail |
| n02132136 | brown bear | head, torso, front_left_leg, front_right_leg, back_left_leg, back_right_leg, and tail |
| n02133161 | american black bear | head, torso, front_left_leg, front_right_leg, back_left_leg, back_right_leg, and tail |
| n02134084 | ice bear | head, torso, front_left_leg, front_right_leg, back_left_leg, back_right_leg, and tail |
| n02134418 | sloth bear | head, torso, front_left_leg, front_right_leg, back_left_leg, back_right_leg, and tail |
| n02389026 | sorrel | head, torso, front_left_leg, front_right_leg, back_left_leg, back_right_leg, and tail |
| n02391049 | zebra | head, torso, front_left_leg, front_right_leg, back_left_leg, back_right_leg, and tail |
| n02397096 | warthog | head, torso, front_left_leg, front_right_leg, back_left_leg, back_right_leg, and tail |
| n02403003 | ox | head, torso, front_left_leg, front_right_leg, back_left_leg, back_right_leg, and tail |
| n02408429 | water buffalo | head, torso, front_left_leg, front_right_leg, back_left_leg, back_right_leg, and tail |
| n02412080 | ram | head, torso, front_left_leg, front_right_leg, back_left_leg, back_right_leg, and tail |
| n02415577 | bighorn | head, torso, front_left_leg, front_right_leg, back_left_leg, back_right_leg, and tail |
| n02417914 | ibex | head, torso, front_left_leg, front_right_leg, back_left_leg, back_right_leg, and tail |
| n02422106 | hartebeest | head, torso, front_left_leg, front_right_leg, back_left_leg, back_right_leg, and tail |
| n02422699 | impala | head, torso, front_left_leg, front_right_leg, back_left_leg, back_right_leg, and tail |
| n02423022 | gazelle | head, torso, front_left_leg, front_right_leg, back_left_leg, back_right_leg, and tail |

**Table 7: Parts taxonomy of DSPart.** †: indicate the left and right parts are separate part classes.

## A  DETAILS ABOUT DSPART

### A.1  PARTS TAXONOMY

Table 7 contains the parts taxonomy for the 50 rigid object classes in DSPart-Rigid and 40 animal classes in DSPart-Animal.

### A.2  VIEWPOINTS SAMPLING RULES

Following (Ma et al., 2024), we sample the object viewpoint with a uniform distribution over the azimuth angle and Gaussian distributions over the elevation and theta angles.

### A.3  ANNOTATOR RECRUITMENT GUIDELINE

To improve the quality of the collected data, each annotator must complete an onboarding stage before starting. The onboarding stage includes training sessions where we present Blender tutorials to teach annotators how to annotate with the help of convenient Blender functions efficiently, and detailed instructions with demo videos on annotating strategies and proper ways to handle edge cases. Additionally, each annotator must annotate three CAD models and meet our standards to qualify for subsequent annotations. We also provided 1 hour of Zoom training sessions and 2 hours of Q&A sessions on handling specific boundary cases which are difficult for annotators.

## B  3D ANIMAL POSE ESTIMATION

Table 8 presents our 3D animal pose estimation results tested on both the real and synthetic test sets of Animal3D (Xu et al., 2023) to demonstrate performance across different domains. The synthetic test set is generated through naive rendering with SMALR (Zuffi et al., 2018) fitted textures. We train the models with our human-filtered animal data for 1000 epochs in the synthetic-only setting. For the synthetic pretrained setting, we pre-train the models on our human-filtered animal data for 100 epochs, then fine-tuned them on real images from the Animal3D training set for 1000 epochs. We use the same metrics as Animal3D (Xu et al., 2023).

The PARE (Kocabas et al., 2021b) methods consistently outperform HMR (Kanazawa et al., 2018) in all settings in terms of 2D PCK, indicating better alignment of predictions with the 2D images. Since 2D part segmentation is often considered complementary to 2D pose estimation (Xia et al., 2017), we select PARE models to filter our animal data. For 3D metrics, PARE showed better performance in terms of S-MPJPE, suggesting more accurate predictions of rigid 3D body poses. However, PARE performs worse in terms of PA-MPJPE, indicating lower accuracy in predicting animal articulations. Therefore, we do not set a strict threshold for PA-MPJPE in our filtering process due to the unreliable predictions.

Despite the presence of some noisy samples with inaccurate 3D poses and articulations in our DSPart-Animal, leading to a performance drop in PA-MPJPE compared to the synthetic pretrained results from Animal3D, we still achieve better performance in S-MPJPE and 2D PCK. We believe our synthetic data pipeline for animals shows promise, but fundamental improvements in conditional synthesis via ControlNet (Zhang et al., 2023a) are necessary.

## C  EXPERIMENTS DETAILS

### C.1  IMPLEMENTATION DETAILS

We disable the Thing-Class ImageNet Feature Distance(FD) (Hoyer et al., 2022) for the UDA method. It is a regularization technique that uses ImageNet features trained from objects to provide guidance to segment object classes, which is inappropriate for segmenting semantics parts of objects. Our goal is to compare baseline performance between other synthetic part datasets with DSPart so we avoid complex data augmentations and leave it for future work to explore the benefits of data augmentation. For all experiments, we only adopt random horizontal flips.

| method | test set | synthetic only | | | synthetic pretrained | | |
|---|---|---|---|---|---|---|---|
| | | PA-MPJPE↓ | S-MPJPE↓ | PCK↑ | PA-MPJPE↓ | S-MPJPE↓ | PCK↑ |
| HMR | Synthetic Animal3D | 155.5 | 715.4 | 45.2 | 143.9 | 565.9 | 54.5 |
| PARE | | 241.5 | 677.9 | 78.1 | 202.9 | 510.8 | 81.2 |
| HMR | Animal3D | 150.8 | 660.1 | 58.2 | 127.0 | 475.2 | 68.9 |
| PARE | | 252.8 | 741.2 | 83.3 | 168.8 | 383.1 | 95.1 |
| HMR* | Animal3D | - | - | - | 124.8 | 497.7 | 63.1 |
| PARE* | | - | - | - | 127.2 | 392.3 | 83.7 |

Table 8: **3D animal pose estimation results.** 'Synthetic Animal3D' refers to direct renderings from fitted SMAL poses and extracted textures. *: results obtained from Animal3D paper that pretrain on the train split of synthetic Animal3D.

| datasets | supervision | architecture | body | tire | mirror | background | mIoU |
|---|---|---|---|---|---|---|---|
| UDAPart | Syn-only | SegFormer | 79.02 | 59.54 | 10.96 | 84.68 | 58.55 |
| | UDA | DAFormer | 86.82 | 55.02 | 28.06 | 92.48 | 65.59 |
| DSPart-Rigid | Syn-only | SegFormer | 82.14 | 61.35 | 1.55 | 87.85 | 58.22 |
| | UDA | DAFormer | 89.47 | 74.34 | 33.75 | 93.11 | 72.67 |
| PartImageNet | Real-only | SegFormer | 92.36 | **79.16** | 29.68 | **95.12** | 74.08 |
| DSPart-Rigid + PartImageNet | Syn+Real | SegFormer* | **92.43** | 78.91 | **34.82** | 95.06 | **75.30** |

Table 9: **Full part segmentation results on car parts.** IoU of each car part is reported. *: models trained with synthetic data and real data successively for each iteration. Numbers are averaged over 3 random seeds.

## C.2 FULL PART RESULTS ON RIGID OBJECT

Table 9 shows the full part segmentation results of each part for the car category, where we observe similar trends as introduced above. We also observe that the lower IoU of the mirror is the reason that leads to a performance drop under the Syn-only setting with DSPart-Rigid. This is possibly because the car CAD models of some car classes we used do not have the mirror parts 3, which leads to a lower pixel distribution of the mirror than UDAPart 10. Additionally, 3D-DST (Ma et al., 2024) has limitations in generating details of small parts, which is another important reason. Notably, DSPart-Rigid significantly enhances the performance of segmentation models on certain challenging part classes (*e.g.*, tire and mirror) under the UDA setting.

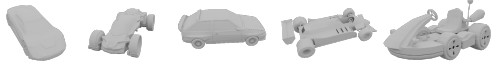

| pixel frequency | mirror |
|---|---|
| UDAPart | 0.116% |
| DSPart-Rigid | 0.055% |

Table 10: Pixel frequency of the mirror in UDAPart and DSPart-Rigid in terms of car categories.

Figure 3: Example of 3D CAD models that do not have mirror parts.

## C.3 ERROR BARS

Table 11 reports the error bars of our main part segmentation results on PartImageNet (He et al., 2022) in the 95% confidence interval. The results are calculated over three runs (random seed 0, 1, 2). We can observe that training with DSPart data will bring higher error bars. This is because DSPart has noisy synthetic images that are hard to filter by model or human (*e.g.* missing small parts, slight 3D pose inconsistency, etc.), and thus the sampling order will influence the results more.

| datasets | supervision | architecture | car | airplane | boat | bicycle | animal |
|---|---|---|---|---|---|---|---|
| UDAPart | Syn-only | SegFormer | $58.55 \pm 1.04$ | $39.10 \pm 1.13$ | N/A | $35.79 \pm 0.64$ | N/A |
| | UDA | DAFormer | $65.59 \pm 1.44$ | $47.73 \pm 1.69$ | N/A | $39.67 \pm 3.43$ | N/A |
| DSPart-Rigid | Syn-only | SegFormer | $58.22 \pm 3.57$ | $58.04 \pm 1.28$ | $61.63 \pm 5.10$ | $37.03 \pm 3.45$ | N/A |
| | UDA | DAFormer | $72.67 \pm 2.10$ | $59.45 \pm 1.72$ | $65.31 \pm 5.74$ | $51.55 \pm 1.47$ | N/A |
| CC-SSL | Syn-only | SegFormer | N/A | N/A | N/A | N/A | $35.46 \pm 2.20$ |
| | UDA | DAFormer | N/A | N/A | N/A | N/A | $47.97 \pm 2.29$ |
| DSPart-Animal | Syn-only | SegFormer | N/A | N/A | N/A | N/A | $44.48 \pm 2.50$ |
| | UDA | DAFormer | N/A | N/A | N/A | N/A | $54.62 \pm 3.66$ |
| PartImageNet | Real-only | SegFormer | $74.08 \pm 2.23$ | $69.98 \pm 2.14$ | $81.71 \pm 1.28$ | $66.27 \pm 1.41$ | $73.16 \pm 0.29$ |
| DSPart-Rigid + PartImageNet | Syn+Real | SegFormer* | $75.30 \pm 0.27$ | $72.43 \pm 2.23$ | $82.51 \pm 2.09$ | $66.70 \pm 1.28$ | N/A |
| DSPart-Animal + PartImageNet | Syn+Real | SegFormer* | N/A | N/A | N/A | N/A | $75.52 \pm 0.40$ |

**Table 11: Error bars of the part segmentation results**. N/A: no classes in that super-category are available in the datasets.

## D  LIMITATION

While 3D-DST (Ma et al., 2024) offers rendered images with free textures and backgrounds, it also introduces issues such as missing details in small object parts and incorrect 3D poses from unusual viewpoints. Although the proposed PRF can alleviate these challenges, a more fundamental improvement in the conditional synthesis via ControlNet (Zhang et al., 2023a) is necessary for future work.

## E  VISUAL COMPARISON OF PRF AND KCF

KCF (3D-DST)

3D rigid pose error (↓):  2.32 < 3.73   **Keep**

KCF (3D-DST)

3D rigid pose error (↓):  1.23 < 3.73   **Keep**

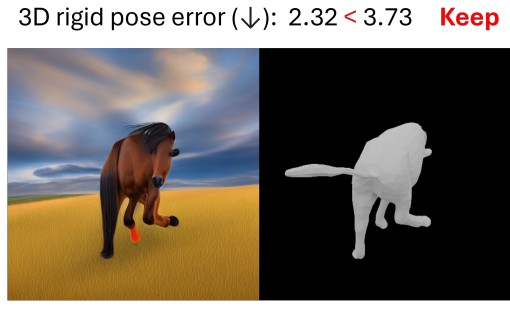
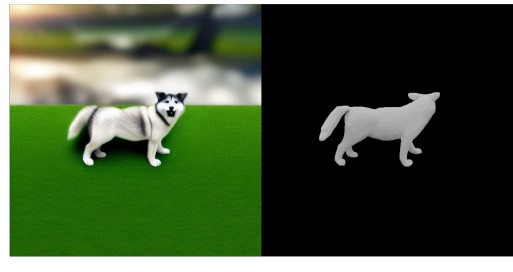

PRF (ours)

PA-MPJPE (↓): 113.79 > 108.06

S-MPJPE (↓): 574.28 > 499.99     **Reject**

2D PCK (↑): 23.08 < 95.83

PRF (ours)

PA-MPJPE (↓): 98.81 < 108.06

S-MPJPE (↓): 531.54 > 499.99     **Reject**

2D PCK (↑): 100.00 > 95.83

**Figure 4:** Examples that are filtered by PRF but kept by KCF. We show the pose error, PA-MPJPE, S-MPJPE, and 2D PCK for each image and show the comparison with the filtering thresholds.

