# OpenReview forum: "DSPart: A Large-scale Diffusion-generated Synthetic Dataset with Annotations from 3D Parts"
_ICLR.cc/2025/Conference — Submitted to ICLR 2025_

### Official Review · Reviewer_yTbV · 2024-10-31

**Soundness:** 2
**Presentation:** 3
**Contribution:** 2
**Rating:** 5
**Confidence:** 4

**Summary:**

This paper proposes two datasets of 3D object parts, DSPart-Rigid and DSPart-Animal. For DSPart-Rigid, it annotates 475 representative shape instances from 50 object categories. For DSPart-Animal, it creates one single 3D part annotation since all SMAL models share the same vertex IDs, and uses 3,065 SMAL models fitted poses from 40 animal categories. It uses a diffusion model, 3D-DST (Ma et al., 2024), to render the realistic images. To further filter out low-quality images for animals, this paper proposes a filter, Part-attention Regressor Filter, which estimates the accuracy of articulations. In total, this data sets contains 48K synthetic images of rigid objects and 52K synthetic images of animals. To evaluate the quality of the dataset, this paper trains the models on its datasets in both synthetic-only and synthetic-to-real scenarios to evaluate their performance on PartImageNet (He et al., 2022) and PascalPart (Chen et al., 2014).

**Strengths:**

1. The proposed filter for animals, Part-attention Regressor Filter, significantly improves the quality of the generated data, as shown in Table 5.

**Weaknesses:**

1. If we compare the results of the model trained only on real-world data and the model trained only on synthetic data, the difference is huge, especially for animals (Table 3 & 4). This might mean that the proposed dataset and dataset creation methods are still far behind the real-world hand data, and might not be scalable as described.
2. Although the proposed filter, PRF, takes a significant portion of the paper, it seems not used in producing the final datasets (see numbers in Table 3 "DSPart-Animal" and Table 5 "Human"). Therefore, it is questionable whether the method is scalable as claimed. I wish it could be further clarified.

**Questions:**

1. In Table 5, it shows that the PRF-filtered result under the UDA setting is close to the human-filtered result, except background. Is there any insights why background is special?
2. It is interesting that, in Table 4 (Syn-only), the IoU of back legs and the tails are significantly worse than the previous synthesized dataset, CC-SSL. Is there any insights on that?

---

> ### Author Response · Authors · 2024-11-23
> **Response to Reviewer yTbV**
>
> >Q1: In Table 5, it shows that the PRF-filtered result under the UDA setting is close to the human-filtered result, except background. Is there any insights why background is special?
>
> >A: In Table 5,  “torso” and  “legs” of human-filtered data are 3.7 and 4.03 higher respectively in terms of IoU. These two parts occupy the most area of the foreground. Although the head IoU is close, and the tail seems to be much better, the performance of the background is still severely affected by the performance of the torso and legs.
>
> >Q2: It is interesting that, in Table 4 (Syn-only), the IoU of back legs and the tails are significantly worse than the previous synthesized dataset, CC-SSL. Is there any insights on that?
>
> >A: It is caused by the limitation of ControlNet and Stable Diffusion in generating small parts and accurate 3D poses from unusual viewpoints (refer to Limitation in the supplementary of current submission).  The tail shapes of DSPart-Animal images are not always accurate and sometimes have distortions. However, CC-SSL was simply rendered by 3D assets which always have well-shaped tails.  For back legs, back-views of animals are often the unusual viewpoints that 3D-DST fails. So DSPart-Animal has a very limited number of back-view animal images, and it will decrease the pixel frequency of the back legs. It also influences the performance of the tail.
>
> >Answer to weakness 1:
>
> >There is still a domain gap between the diffusion-generated data and real data. However, we demonstrate that DSPart is closer to real data distribution through experiments on PartImageNet and PascalPart than other synthetic datasets. Currently, the problem in terms of scalability is more caused by the bias of ControlNet and Stable Diffusion which leads to a high failure rate in generating images of objects with more complex structures from unusual viewpoints.
>
> >Although it will introduce a more severe viewpoint bias, a more scalable way to create animal images (where there is a high failure rate) is to follow the viewpoints from the human-filtered data. The generation pipeline will be more comfortable with those viewpoints and they can be applied to most 3D objects to generate more reliable samples. However, for DSPart, we want to create a viewpoint space that is as large as possible for our synthetic data. Although most unusual viewpoints are filtered by KCF or PRF, we can still obtain some successful examples for certain 3d objects under unusual viewpoints.
>
> >Answer to weakness 2:
>
> >The results shown in Table 3 are obtained using the 13k human-filtered data to achieve the highest performance. The PRF results in Table 5 are the results of 13k filtered images from other 150k unfiltered samples. For the left 26k images in DSPart-Animal, we are using more strict thresholds and we sample these 26k images from more than 300k unfiltered samples. We report the results of these 26k images which we use to evaluate the filtering quality here:
>
>  | supervsion |  head |  torso | leg |  tail | bg |  mIoU |
> |:-----------:|:-----:|:-----:|:------:|:-----:|:----------:|:-----:|
>  |   syn-only  | 46.52| 31.90 |  21.88  |  10.78 |   81.95   | 38.61 |
>  |     UDA     | 54.37 | 52.05 |  47.04  |  17.13 |   93.26  | 52.77 |
>
> >We will make these clearer in the revision and add the results of the unused DSPart-Animal images in the supplementary.

---

> > ### Comment · Reviewer_yTbV · 2024-11-24
> >
> > Thanks for the response. In this case, it is still questionable whether the method is indeed scalable. This paper would be much stronger if the authors can show how many images are needed for the proposed filter to beat human-filter. It could be even better if there is a relationship plot for them to achieve the same performance, i.e., some scaling law for generative dataset.

---

### Official Review · Reviewer_hvMi · 2024-10-31

**Soundness:** 3
**Presentation:** 3
**Contribution:** 3
**Rating:** 6
**Confidence:** 4

**Summary:**

The paper introduces a synthetic dataset called DSPart, designed to improve part-based recognition models. DSPart includes synthetic images with part-level annotations derived from 3D models, specifically addressing two key groups: rigid objects (DSPart-Rigid) and animals (DSPart-Animal). With annotating 3.5K 3D models, it generates 100K realistic synthetic images with masks. Extensive experiments demonstrate that DSPart surpasses existing synthetic datasets in part segmentation tasks.

**Strengths:**

1. The paper introduces an innovative approach that utilizes 3D part annotations to generate realistic synthetic images with corresponding part masks. This method enables the generation of an unlimited number of synthetic data samples using a limited set of annotated 3D shapes, significantly reducing the cost and time required for annotating large quantities of 2D images.
2. Sufficient experiments on the 2D part segmentation task to demonstrate the effectiveness of the DSPart dataset.
3. DSPart addresses limitations in previous synthetic datasets by using diffusion models to generate realistic textures and physically consistent contexts, making the synthetic images more reflective of real-world conditions.

**Weaknesses:**

1. The method is challenging to generalize due to the difficulty in defining annotations for parts in other categories. For instance, providing semantic-level part definitions for data in the large-scale 3D dataset Objaverse[1] is particularly challenging. Is it able to consider a part definition approach similar to SAM?
2. Generating images with 2D part annotations using standard 3D parts can increase data volume; however, annotating 3D data is more challenging and requires annotators to possess advanced tool usage skills. This, in turn, limits the scalability of the data to some extent.
3. Each generated image contains a single, clear object positioned in the center. However, in real images, there may be multiple objects that are not necessarily centered or clearly defined. This indicates that some domain gaps still exist.

[1] Deitke M, Schwenk D, Salvador J, et al. Objaverse: A universe of annotated 3d objects[C]//Proceedings of the IEEE/CVF Conference on Computer Vision and Pattern Recognition. 2023: 13142-13153.

**Questions:**

1. Is it able to adjust poses of 3D shapes to increase the diversity of 3D data?

---

> ### Author Response · Authors · 2024-11-23
> **Response to Reviewer hvMi**
>
> We thank the reviewer for their valuable feedback.
>
> >Q1. Is it able to adjust poses of 3D shapes to increase the diversity of 3D data?
>
> >A:  Generally, we can adjust the 3D rotation, i.e. SO(3), and also some translation that won’t make the object truncated.  For articulated objects like animals, we can create more animal poses by adjusting certain pose parameters of SMAL to change the joint angles. We can start with the SMAL poses and viewpoints combinations from the human-filtered data to guarantee the accuracy of generated samples. However, the difficulties lie in not making impossible and unrealistic animal poses when manipulating.
>
>
> >Answer to **Weakness 1**:
>
> >We agree that it is difficult to define part for large-scale categories. We think DSPart can consider the part definition approach of SAM. Our current 3d part definitions are adding spatial information to surface-based parts (closer to Pascal-Part style) since we don’t want annotating our defined parts in real images to be nearly impossible. These surface-based parts also enable us to leverage SAM.  We can render the 3D assets from Objaverse into images from several viewpoints so that all parts can be visible, and then use SAM to give a subregion-based part segmentation. Then we can add our extra spatial information to these SAM-generate parts. Note that if two parts are both in one continuous and smooth surface, SAM often cannot segment the boundary (e.g. airplane head and body). Therefore, we still extra efforts but SAM can make the scaling up of part definitions much easier.
>
> >Answer to **Weakness 2**:
>
> >(1) Annotating 3D data is challenging but our annotation protocol only requires basic navigation operations of Blender that can be taught quickly in a one-hour session to annotators.
>
> >(2) It indeed limits the scalability but the scalability is still much better than annotating 2D parts in more fine-grained part definitions. A lot of self-occlusion, low resolution for some images, and natural shading will make annotating fine-grained parts in real images nearly impossible. SAM will not work well even with accurate bounding boxes when dealing with fine-grained parts.
>
>
> >Answer to **Weakness 3**:
>
> >We can generate the aforementioned images by utilizing multi-control, adding translations, and introducing occluders. However, the current design of DSPart aims to address challenging object-level part problems (e.g., 2D-3D part correspondences). Therefore, we are creating datasets in simpler scenarios to focus exclusively on our targeted challenges.

---

> > ### Comment · Reviewer_hvMi · 2024-11-25
> >
> > Thanks for your reply !
> >
> > I also believe that the part definition of SAM is a good idea. I hope the paper can discuss such part definition in the future version. I will keep my score.

---

### Official Review · Reviewer_xgjX · 2024-11-02

**Soundness:** 2
**Presentation:** 2
**Contribution:** 2
**Rating:** 5
**Confidence:** 4

**Summary:**

This paper presents a new part-segmentation synthetic dataset via 2D diffusion models, the results show that it is possible to use diffusion models to generate more diverse data samples for part segmentation task.

**Strengths:**

1. Show the potentional on using synthetic data for part segmentation.

**Weaknesses:**

1. Lack of details on how to use diffusion models to generate data samples, which base model is used. Why not use a ControlNet to generate controllable data pairs.

2. The scale of the generated data is relatively small, which makes it hard to verify the generalization capability of the model when trained on  larger data scales.

**Questions:**

1. Given the categories included in DSPart-Rigid and DSPart-Animal, to what extent is this dataset likely to be generalizable to other object categories or part segmentation tasks?

---

> ### Author Response · Authors · 2024-11-23
> **Response to Reviewer xgjX**
>
> >Q1: Given the categories included in DSPart-Rigid and DSPart-Animal, to what extent is this dataset likely to be generalizable to other object categories or part segmentation tasks?
>
> >A:  In 2d part segmentation,  the dataset can be generalizable to other object categories that are able to share the same part definitions as the available ones.
>
> >For potential 3d tasks that can leverage DSPart,  we are confident that the 3d parts in DSPart can construct reasonably good part shape space that can generalize to novel shapes. We select our 3d assets that have enough shape variance within each category from large-scale 3d datasets (ShapeNet [1] and Objaverse [2]), making it possible to construct a promising shape pace of that category by interpolation of our selected shapes. Since parts have much simpler shapes (i.e. cuboid-like, cylindrical, spherical, etc), they can be represented by the interpolation of limited representative samples.
>
>
> >Answer to **weakness 1**:
>
> >We use Stable Diffusion 1.5 and control_v11p_sd15_canny for controlNet.
>
> >For the controllable data pairs, we believe the 3D-DST pipeline we employed exactly uses ControlNet to generate such pairs. Specifically, for each 3D asset, we render it from various viewpoints and utilize the edge maps of the rendered images to generate synthetic images. The 3D control is achieved by using the edge maps of different rendering configurations.
>
>
> >Answer to **weakness 2**:
>
> >The pipeline can be applied to any existing 3D shape part dataset to generate image and annotation pairs to scale up.  However, the problem is the lack of annotated real datasets for evaluation.
>
> **Reference**
>
> [1] Chang et al. - ShapeNet: An Information-Rich 3D Model Repository.
>
> [2] Deitke et al. - Objaverse: A Universe of Annotated 3D Objects.

---

### Official Review · Reviewer_qEFu · 2024-11-04

**Soundness:** 3
**Presentation:** 4
**Contribution:** 2
**Rating:** 5
**Confidence:** 3

**Summary:**

The paper introduces a framework (DSPart), including a filtering approach (PRF), for generating a large-scale synthetic dataset for 2D object part segmentation (DSPart-Rigid and DSPart-Animal). The results demonstrate that training on the generated dataset achieves superior performance compared to existing synthetic datasets (UDAPart and CC-SSL), in both supervised (SegFormer) and unsupervised (DAFormer) setups.

**Strengths:**

1.	The paper is well-written and easy to follow.

2.	The paper presents an effective and easily applicable method (DSPart) for utilizing an existing generation approach (3D-DST) and existing 3D CAD models (e.g., ShapeNet and Objaverse), along with additional annotations (e.g., 3D parts), to generate pixel-level part-based datasets.

3.	The scale and diversity of the generated dataset (DSPart-Rigid and DSPart-Animal) are impressive, comprising 100k images, 90 object categories, and over 3.5k 3D models.

4.	Training on the generated dataset, whether or not UDA is applied, typically demonstrates superior results on real part-based datasets (e.g., PartImageNet) compared to training on existing synthetic datasets.

**Weaknesses:**

Major:

1.	Despite its simplicity, the proposed framework (DSPart) shows limited novelty from a technical perspective. It appears more like a direct application of an existing framework (3D-DST), enhanced by incorporating additional annotations (e.g., 3D parts) and a wider variety of 3D models (e.g., animals). Although the paper claims that the extension is non-trivial (Line 107), it would be beneficial if the paper could justify this claim by highlighting its technical innovations.

2.	The paper demonstrates that the proposed filtering mechanism (PRF) outperforms the existing one (KCF). However, this comparison seems unfair and potentially lacks meaningful insight for several reasons. First, PRF trains models for filtering using reliable human-filtered data (e.g., 13k images) that KCF does not have access to, making it unsurprising that PRF performs better. Additionally, the paper does not quantify the contribution of each component within PRF, such as the use of two PARE models, three different metrics, and various thresholds. It is possible that PRF’s superior performance is simply due to its access to more reliable data. Finally, the paper does not provide visual examples to illustrate why PRF outperforms KCF. Including examples of data filtered by PRF but not by KCF, or showcasing PRF's failure cases, would add clarity and strengthen the explanation.

3.	Although the paper shows that using the DSPart dataset for training generally yields better performance than other existing synthetic datasets, it does not fully explain why the DSPart dataset is superior, which is essential for the community to understand the results. For example, as shown in Table 2, training models on DSPart-Rigid consistently outperforms training models on UDAPart. Is this advantage due to the realism of the generated images, the diversity of the 3D CAD models, the increased number of viewpoints, or the total number of generated images? Investigating different versions of DSPart-Rigid could be insightful, such as generating DSPart-Rigid with the same set of 3D CAD models used by UDAPart, varying the number of viewpoints, generating different numbers of images, or controlling texture diversity through varied prompts.

Minor:

4.	In Table 5, the paper shows that PRF is more efficient (20 minutes for inference) compared to KCF (50 training hours). However, since PRF requires human-filtered data for training 3D pose estimation models, it would be fairer to include this as part of PRF’s overall cost (e.g., 100 hours human labor). While the paper mentions that PRF only needs a pre-trained model and can be applied to the generated images at scale, this is true only if the human-filtered data used by PRF effectively captures the entire animal pose space, which may not be the case.

5.	The paper defines five super-categories (car, airplane, bicycle, boat, and tool) for rigid objects in the proposed dataset (DSPart-Rigid). However, the tool category is never used in the paper.

6.	The paper utilizes only a single architecture for both the fully supervised setup (SegFormer) and the UDA setup (DAFormer). Including additional architectures or UDA methods would strengthen some of the paper’s claims, such as the claim that DSPart-Rigid enables better knowledge transfer from the synthetic to the real domain for UDA (Lines 355-357).

**Questions:**

1.	Please clarify the technical novelty of the proposed framework in response to Weaknesses-1. Highlighting the key advancements distinguishing 3D-DST from the proposed framework would be beneficial.

2.	Please provide additional details about PRF in response to Weaknesses-2 and Weaknesses-4. This may include explaining why PRF outperforming KCF is still meaningful, even if PRF has access to more reliable data, quantifying the contribution of each component within PRF, and providing visual examples.

3.	Please fully justify why the DSPart dataset is superior. As suggested in Weaknesses-3, exploring different versions of DSPart-Rigid could be insightful.

---

> ### Author Response · Authors · 2024-11-23
> **Response to Reviewer qEFu**
>
> >Q1. Please clarify the technical novelty of the proposed framework in response to Weaknesses-1. Highlighting the key advancements distinguishing 3D-DST from the proposed framework would be beneficial.
>
> >A: We thank the reviewer for their feedback and would like to clarify the main contributions of our work, which lie in three key aspects:
> - Proposed PRF Filtering Strategy:
>
> Our PRF strategy introduces a novel filtering approach that goes beyond 3D rigid body poses to incorporate articulation accuracy. By utilizing 2D and 3D part annotations in the DSPart dataset, we replace the 3D rigid rotation (SO(3)) metrics from KCF in 3D-DST with a combination of PA-MPJEP, S-MPJPE, and 2D PCK. This approach achieves more accurate filtering and avoids the vulnerability to unreliable associated with K-fold splits used in 3D-DST. Importantly, we are the first to apply 3D animal pose estimation for filtering diffusion-generated synthetic animal images. While this requires approximately 100 hours of human annotation, it also ensures a reliable and high-quality training subset for fine-tuning scenarios.
> - Dataset Features and Annotation:
>
> For a dataset paper, we argue that the form of data and annotation is also a crucial novelty. Compared to previous synthetic datasets, our DSPart dataset provides several key advantages:
>
> (1) **Realistic textures and plausible backgrounds for the generated images.**
>
> (2) **Accurate camera parameters, 2D part masks, and 3D part geometry for each image.**
>
> As a comparison, among these attributes, 3D-DST satisfies only the first point and 3DCoMPaT++ satisfies only the second point. To the best of our knowledge, no existing synthetic datasets combine both features.
> - Dataset Design:
>
> The ways of shape selection and part definitions in DSPart are novel and meaningful for potential research. The assets of DSPart-Rigid are selected to have enough shape variance within each category since we want the shape space of that category to be represented by interpolation of our selected shapes. That shows a promising future in using our 3d parts to construct part-shape spaces. Unlike other fine-grained 3d part datasets that include many small parts and parts in the object interior which are nearly impossible to annotate in real images and too difficult for the current model, our 3D part annotations are fine-grained with adding spatial information to surface-based parts (more close to Pascal-Part style). Thus annotating real images for evaluation will be relatively more approachable. Adding spatial information can also motivate research to solve existing 3D spatial awareness problems (i.e. spatial ambiguity) of current algorithms.
>
> In conclusion, we argue that DSPart is novel as a synthetic dataset. The filtering strategy has limited technical novelty, but as a dataset paper, we think it is not the most important component of this paper. The unique features of DSPart are designed to address current challenges and inspire new 2D-3D tasks in computer vision. We hope the reviewer will reconsider our contributions in light of these points.

---

> ### Author Response · Authors · 2024-11-23
> **Response to Reviewer qEFu - continued**
>
> >Q2: Please provide additional details about PRF in response to Weaknesses-2 and Weaknesses-4. This may include explaining why PRF outperforming KCF is still meaningful, even if PRF has access to more reliable data, quantifying the contribution of each component within PRF, and providing visual examples.
>
> >A: Given weakness 2, we train the KCF on the same human-filtered data as PRF in two settings: 1) For each fold, we train the 3D pose estimation model on the human-filtered data for 50 epochs and then train on the train split of this fold for the left 50 epochs  2)  We train the 3D pose estimation model on the human-filtered data for 100 epochs and directly evaluate it on the unfiltered data (the same process as PRF).  The 2nd setting can prevent the model from training on less reliable data although it could not be called K-fold anymore since we do not perform the train/val splits on unfiltered data.  The experiment results are shown below:
>
> | data | supervsion |  head |  torso | leg |  tail | bg |  mIoU |
> |:---------:|:-----------:|:-----:|:-----:|:------:|:-----:|:----------:|:-----:|
> | KCF 1st |   syn-only  | 22.60 | 20.85 |  18.28  |  12.33 |    76.55   | 30.12 |
> | KCF 1st |     UDA     | 20.74 | 31.96 |  44.42  |  30.63 |    85.03    | 42.56 |
> | KCF 2nd |   syn-only  | 17.22 | 22.39 |  20.25  | 13.57 |    75.47    |  29.78 |
> | KCF 2nd |     UDA     | 19.81 | 31.22 |  46.09 |  21.40 |    85.02   | 40.71 |
>
> Surprisingly, training only on reliable data (KCF 2nd) will not obtain better-filtering results based on 3D rigid rotation. That the UDA performance of KCF 1st is worse than the original KCF in Table 5 further demonstrates it. We provide a visual comparison of examples that are filtered by PRF but not by KCF with their corresponding metrics and thresholds (please refer to Figure 4 in the supplementary of revision).
>
>
> For weakness 4, we agree that current ways of comparing efficiency are misleading, which will be revised and clarified in the revision. The KCF’s inefficiency lies in its K-fold design which requires several train-test splits for every bunch of data. A fair comparison is to compare PRF with the 2nd setting above, which directly demonstrates the meaningfulness of PRF in changing metrics and algorithms to 3D animal pose estimation. The human-filtered data is also an important contribution of the PRF pipeline. It can be severed as more reliable data for fine-tuning or testing purposes.
>
> For the doubt in scalability, the pose space indeed contains bias for future filtering. It is mainly caused by the bias of ControlNet and Stable Diffusion, which leads to a high failure rate in generating images of some animal poses from unusual viewpoints. However, at the performance level, our filtered images by PRF can achieve close performance as the human-filtered data, which is enough for scaling up the data volume following our image generation pipeline.
>
> >Q3: Please fully justify why the DSPart dataset is superior. As suggested in Weaknesses-3, exploring different versions of DSPart-Rigid could be insightful.
>
> >A: We perform the following experiments on the car category for further illustration.
> |            data       | # CAD models |  # viewpoints |  # images | syn-only |  UDA |
> |:-----------------------:|:-----------:|:-----:|:-----:|:------:|:-----:|
> |                   UDAPart         |   5  | 500 | 2500 |  58.40  |  64.71 |
> |                   DS-UDAPart        |    5   | 500 | 2500 |  58.89  |  69.27 |
> |               DS-UDAPart             |   5  | 500 | 5000 |   59.14  | 70.41 |
> |               DS-UDAPart              |    5    | 500  | 10000 |  59.01 |  70.53 |
> |               DS-UDAPart              |     5    | 800* | 4000 |  59.34 |  71.64 |
>
> Note that the viewpoints are uniformly sampled from the same interval of azimuth and elevation angles. The 5 CAD models of car categories are not included in DSPart-Rigid and we generate another set of images called DS-UDAPart. To control the viewpoints to be the same, we choose not to filter the images, and thus there will be some synthetic images with inaccurate annotations under unusual viewpoints.
>
> The comparison between row 1 and row 2 implies the benefit of the realism of generated images.  The comparison between row 2 and row 3 implies the benefit of increased data volume. However, when the amount of data grows beyond a certain point, performance improvements become increasingly marginal because the improvement brought by diverse textures has already saturated (row 3 & 4).

---

> > ### Author Response · Authors · 2024-11-23
> > **Continued (part 2)**
> >
> > >Continuation of Answer to Q3.
> >
> > In row 5, we add 300 additional different viewpoints. To mitigate the influence of the addition of more failure cases, we sample 500 different viewpoints from the previous 500 viewpoints. We perform KCF on 500 images of one CAD model and filter 300 viewpoints. For the other 4 CAD models, we directly choose these 300 viewpoints.  It is hard to separate the effects of the number of viewpoints and the number of images in our setting. Adding more viewpoints means more images and thus means potentially more diverse textures will be introduced. But by comparing the results of row 2, 3, and 5, more viewpoints may have a positive effect.
> >
> > Additionally, the performance of DS-UDAPart of only 5 CAD models is close to the performance of DSPart-Rigid generated with many more 3D shapes. For 2D part segmentation on PartImaegNet where the part definitions of cars are too coarse, the influence of the diversity of the 3D CAD models is hard to access. Only several representative shapes like UDAPart will be enough to get reasonably good performance on PartImageNet.
> >
> > Currently, it is hard to guarantee the generated image is consistent with its textual prompts. Our filtering strategy is to maintain the consistency of visual prompts so that we can have accurate annotations. Therefore, it is hard to dig further into controlling texture diversity through varied prompts since the controllability is not guaranteed.
> >
> >
> > > Answer to Weakness 5:
> >
> > The reason we do not evaluate the tool category is because of the lack of real datasets or comparable synthetic data.  Most image part datasets do not have object classes from our tool category (i.e. PartImageNet, PascalPart).  PACO [1] has some similar classes (e.g. bench, broom, hammer, and laptop). The part definitions are different but we could merge for evaluation. However, it is hard to find other synthetic datasets of these classes that can be compared with our synthetic images. Currently, only 3DCoMPaT++ [2] has bench class that can be used for training, but their images are rendered with white ground and only 8 viewpoints. So we do not include evaluation of the tool category in the current submission.
> >
> > > Answer to Weakness 6:
> >
> > According to the benchmark of GTA5 to Cityscapes, most SOTA UDA segmentation methods are based on DAFormer architecture [3, 4, 5, 6, 7] except for some active learning methods. So we think it would be sufficient to test the UDA performance only on DAFormer. After removing the self-training components, the DAFormer becomes SegFormer and thus we use it to evaluate the synthetic-only setting.
> >
> > **Reference**
> >
> > [1] Ramanathan et al. - PACO: Parts and Attributes of Common Objects
> >
> > [2] Slim et al.  - 3DCOMPAT++: An improved Large-scale 3D Vision Dataset for Compositional Recognition.
> >
> > [3] Chen et al. - Transferring to Real-World Layouts: A Depth-aware Framework for Scene Adaptation.
> >
> > [4] Hoyer et al. - MIC: Masked Image Consistency for Context-Enhanced Domain Adaptation.
> >
> > [5] Hoyer et al. - HRDA: Context-Aware High-Resolution Domain-Adaptive Semantic Segmentation.
> >
> > [6] Chen et al. - PiPa: Pixel- and Patch-wise Self-supervised Learning for Domain Adaptative Semantic Segmentation.
> >
> > [7] Xie et al. - SePiCo: Semantic-Guided Pixel Contrast for Domain Adaptive Semantic Segmentation.

---

### Meta-Review · Area_Chair_PicK · 2024-12-07

**Metareview:**

**Summary**

The paper presents DSPart, a dataset of Diffusion-generated Synthetic Parts, consisting of 100K images with part masks generated from 3D models.  Part annotations are provided on the 3D models and used to render part segmentations that corresponds to the realist images generated using 3D-DST [Ma et al 2024].

The DSPart dataset is composed of DSPart-Rigid (48K images generated from 475 rigid 3D models over 50 categories) and DSPart-Animal (52K images generated from 3,065 models from Animal3D that are created by taking the SMAL model and fitting it to 40 animal categories).  The 475 rigid 3D models are annotated with fine-grained parts. For DSPart-Animal, a single part annotation for the SMAL model is sufficient, but a filtering phase is needed to filter out low quality generated images.

Experiments compare whether using the proposed dataset to train part segmentation network can improve performance.  Results show that combining the proposed dataset with existing training dataset can increase accuracy.

**Contributions**
- The main contribution of the work is the proposed method of generating aligned natural images and pixel-level part segmentation using annotated 3D models and 3D-DST.
- The work build on existing components, with the main novel component being the proposed filtering mechanism

**Strengths**
1. Paper is well-written and easy to follow [qEFu]
2. Proposed method is effective at providing pixel-level part segmentation (given part annotated 3D models) [qEFu,hvMi]
3. Experiments demonstrates that including the dataset in training can help improve part segmentation performance [qEFu,xgjX,hvMi]
4. Using generated data for training part segmentators can potentially provide unlimited data [hvMi]
5. Some reviewers felt the scale and diversity of the generated dataset was sufficient [qEFu]

**Weaknesses**

Reviewers expressed the following concerns:
1. Concerns about the scalability of the approach
   - Some reviewers find the scale of generated data to be small [xgjX]
   - Generalizability to other categories unclear [xgjX]
   - The proposed method relies on having annotated parts on 3D models which may be hard to obtain [hvMi]
   - As annotating 3D parts is more challenging than annotating 2D parts, there is doubts about the scalability of the approach [hvMi]
   - Results showing the effectiveness of the DSPart data is with human filter (not the proposed filter mechanism) [yTbV]
2. Concerns that the generated images are not of sufficient quality as real-world images
   - Performance on model trained on real images is still much better than using synthetic images [yTbV]
   - The generated images do not reflect objects in context which are more natural [hvMi]
3. Other concerns:
   - Limited technical contribution, as the work is largely based on 3D-DST. [qEFu]
   - Concerns about fairness of comparison of the proposed filtering mechanism [qEFu]
   - Lack of insight about what aspect of DSPart provides the most benefit to guide how to scale up the generation [qEFum,xgjX]

**Recommendation**

While reviewers appreciated the potential of using generated data to train better part segmentation models, most reviewers did not find the contributions to be sufficient for acceptance at ICLR (with 3 reviewers giving the paper a rating of 5, and one a rating of 6).

The main concern is the scalability of the proposed approach, as it requires annotated 3D parts, and human filtering of images was used for the final dataset.  Due to the concerns expressed by the reviewers, the AC recommends reject.

**Additional Comments On Reviewer Discussion:**

Overall reviewer opinions were unchanged.

Reviewer qEFu provided the most detailed review, but did not engage in discussion with the authors.  Nevertheless, the author response alleviated some of their concerns and they increased their rating from 3 to 5, but remained negative.

Reviewers yTbV and xgjX acknowledge the author's responses to their questions, but still had concerns about the scalability of the proposed generation pipeline.  Reviewer xgjX felt that it was not clear what "key factors are most critical for scaling up the pipeline", while reviewer yTbV had questions about whether the human filtering can be replaced by the proposed automatic filtering mechanism.

Reviewer hvMi, who was slightly positive, remained slightly positive.  However, they also expressed doubts about the scalability of the proposed generation method due to challenges in obtaining 3D part segmentation.

---

### Decision · Program_Chairs · 2025-01-22

Reject